# Dendritic Solidification and Physical Properties of Co-4.54%Sn Alloy with Broad Mushy Zone

**Weili Wang \*, Wenhui Li and Ao Wang**

Department of Applied Physics, Northwestern Polytechnical University, Xi'an 710072, China
\* Correspondence: wlwang@nwpu.edu.cn

**Abstract:** The high undercooling of binary Co-4.54%Sn alloy has a significant influence on its microstructural characteristics and physical properties. Here, we report that the dendritic growth and physical properties of broad-temperature-range Co-4.54%Sn alloy remarkably depends on the undercooling during the rapid solidification. The maximum undercooling attains 208 K at molten state, and the dendritic growth velocity is quite sluggish in highly undercooled liquid Co-4.54%Sn alloy because it has a broad solidification range of 375 K ($0.21\ T_L$); the maximum value is only 0.95 m/s at the undercooling of 175 K, which then decreases with undercooling. The microstructure refines visibly and the volume fraction of the interdendritic $\beta Co_3Sn_2$ phase obviously decreases with undercooling. The microhardness and electrical resistivity increase with undercooling owing to the enhancement of solute content of the primary $\alpha Co$ phase and refinement of the microstructure where the increased crystal boundary hinders the electronic transmission. Meanwhile, the saturation magnetization also reduces with undercooling due to the crystal particle and boundary increasing significantly, and the dendritic growth velocity and solute content increase in the primary $\alpha Co$ phase under rapid solidification.

**Keywords:** rapid solidification; dendritic growth; microhardness; electrical resistivity; magnetic property

## 1. Introduction

The broad solidification temperature range is the temperature range of a solid–liquid two-phase mixture which is larger than $0.2\ T_L$, in which the alloy ingot is in a semisolid state for this liquid/solid range during the solidification; this range is called the broad mushy zone [1,2]. For a binary single-phase alloy, a solidification nucleates from the liquid to solid phase over a broad temperature interval, and then grows rapidly and forms a single-phase microstructure for a considerable time in this mushy zone [2]. Therefore, the phenomenon of solidification shrinkage and hot tearing is caused by thermal contraction, as the higher density of the solid driving the cracking process and hindered liquid flow prevents any void from being filled with liquid. Several defects, such as low fluidity, shrinkage porosity, heat cracking, segregation, and so on, take place inside this interval [1–5]. The rapid solidification of high undercooling is an effective method to avoid or eliminate these defects.

The rapid solidification of metallic alloys under high undercooling has the advantages of extended solubility, refined microstructures and the formation of a metastable phase [6–11]; thus, it can reduce defects, effectively improve microstructure and achieve excellent physical properties. Some binary single-phase alloys with a broad solidification temperature range show special phenomena during the rapid solidification of high undercooling; for example, dendritic growth velocity decreases when the undercooling exceeds a critical value, and the final microstructure is vermicular dendrite instead of equiaxed grains. Moreover, the undercooling and cooling rates have significant influence on the microstructure, microhardness and electrical resistivity properties during the rapid solidification [12,13].

Co-Sn alloys have been widely studied in recent years owing to their application properties; for example, the production of Co-Sn-based metallic glasses and anode materials in lithium ion batteries [14–16]. In addition, the investigation of Co-Sn alloys have also focused on the structural transformation of different Sn content or eutectic alloys [17], physical properties of the viscous flow, surface tension and enthalpy of mixing near the eutectic concentration [18,19]. However, there are scarce studies on the structural evolution, dendritic growth and physical properties of binary Co-Sn alloys with a broad solidification temperature range and rapid solidification at high undercooling.

In this work, we selected a special composition of Co-4.54%Sn alloy with a broad solidification temperature range of 375 K (0.21 $T_L$) to explore the microstructural evolution and dendritic growth characteristics under substantial conditions of undercooling. Moreover, the physical properties of the microhardness, electrical resistivity and magnetic characteristics are investigated at different undercoolings.

## 2. Experimental Procedure

The rapid solidification experiments were performed using the glass fluxing method with a high vacuum chamber. The master alloy of binary Co-4.54%Sn alloy composition was prepared using the arc-melting technique from component metals of high purity above 99.999%. An in situ alloying procedure was applied to prepare about 1 g Co-4.54%Sn alloy using radio frequency induction heating. The sample was contained in an 8 mm ID × 10 mm OD × 12 mm alumina crucible and placed into the experimental chamber, then it was evacuated to a $2 \times 10^{-4}$ Pa vacuum with a turbo pump and subsequently backfilled with argon gas to $10^5$ Pa. The sample was superheated to 250~350 K above its liquidus temperature and kept submerged in a pool of molten fluxing agent for 3~5 min. It was naturally cooled down and solidified by switching off the induction heating power. Each sample was subjected to the melting–solidifying cycle 3~5 times. Their heating and cooling curves were recorded using a Land NQO8/15C infrared pyrometer (Sensortherm Gmbh, Steinbach, Germany), while the dendrite growth velocity was measured with an infrared photodiode device (Mikrotron, Unterschleißheim, Germany). The phase constitutions of binary Co-4.54%Sn alloy were analyzed by Rigaku D/max 2500 X-ray diffractometer (XRD) (Rigaku, Tokyo, Japan), whereas their solidification structures and solute distribution profiles were investigated with FEI Sirion electron microscope (SEM) (FEI Sirion, the Netherlands) and INCA Energy 300 energy-dispersive spectrometer (EDS) (FEI Sirion, the Netherlands). The microhardness was measured using an HXD-2000TMC/LCD Vickers hardness tester (Kexin, Beijing, China). The magnetic characteristics was analyzed using a VSM instrument. The resistivity was tested using the four-point probe method.

## 3. Results and Discussion

### 3.1. Rapid Solidification Structures

The phase constitution of Co-4.54%Sn alloy was selected with the largest solid solubility point of Sn solute in Co solvent; as shown in the binary Co-Sn phase diagram of Figure 1a,b, the liquidus and solidus temperatures were 1760 and 1385 K, respectively, and the solidification temperature range attained 375 K (0.21 $T_L$). From the equilibrium phase diagram, it can be seen that the primary face-centered cubic αCo solid solute phase formed in the liquid phase when the temperature reduced to 1760 K; firstly, during the solidification, the solute content was about 0.56 wt.%Sn in the αCo phase, and then the βCo$_3$Sn$_2$ intermetallic phase was generated as the temperature lowered to about 1443 K. Subsequently, the close-packed hexagonal εCo phase had the potential to form once the temperature decreased to about 695 K. Furthermore, the X-ray diffraction patterns showed three phases of αCo, βCo$_3$Sn$_2$ and εCo displayed at the different undercoolings under the rapid solidification, which corresponds with the equilibrium phase diagram. The microstructures of binary Co-4.54%Sn alloy relate to the undercooling at rapid solidification, which are shown in Figure 2a–d. At the small undercooling of 11 K, the microstructure of the primary αCo phase was characterized by the coarse and well-developed dendrites, the

βCo₃Sn₂ phase distributed into the interdendrites, shown as the white phase and the εCo phase displayed the dark structures, as seen in Figure 2a. Subsequently, the microstructure was remarkably refined and the dendritic feature disappeared gradually; the volume fraction of βCo₃Sn₂ phase decreased at the same time, but the volume fraction of the εCo phase increased with the increase of undercooling. Ultimately, the microstructure of the primary αCo phase grew with the disorder at the largest undercooling of 189 K, and a small number of βCo₃Sn₂ particles and a large number of εCo phase distributed randomly into the matrix of the primary αCo phase. Clearly, the phase constitution has not changed at the rapidly solidified condition. It is worth noting that the eutectic-like microstructure exists in the whole undercooling of the binary Co-4.54%Sn alloy, as seen in Figure 3a,b, which grew around the βCo₃Sn₂ phase. Meanwhile, the eutectic-like structure appears coarsening and the volume fraction increased obviously with increase of undercooling and decrease of the βCo₃Sn₂ volume fraction, and formed eutectic cell characteristics at the largest undercooling of 189 K.

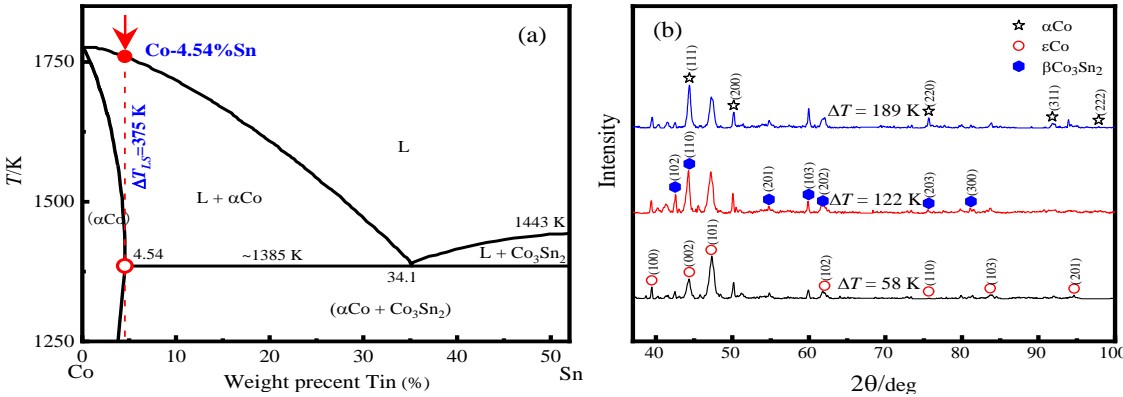

**Figure 1.** Selection of alloy composition and X-ray diffraction of rapidly solidified alloy droplets. (**a**) Co-4.54%Sn alloy location in phase diagram and (**b**) XRD pattern.

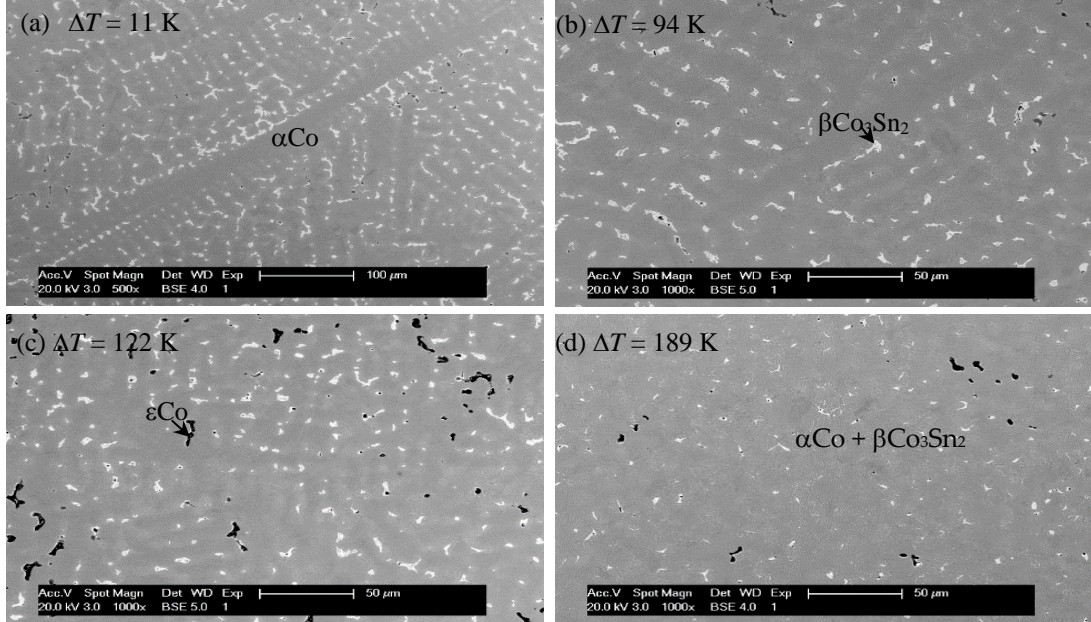

**Figure 2.** Microstructures at different undercoolings, (**a**) 11 K, (**b**) 94 K, (**c**) 122 K, and (**d**) 189 K.

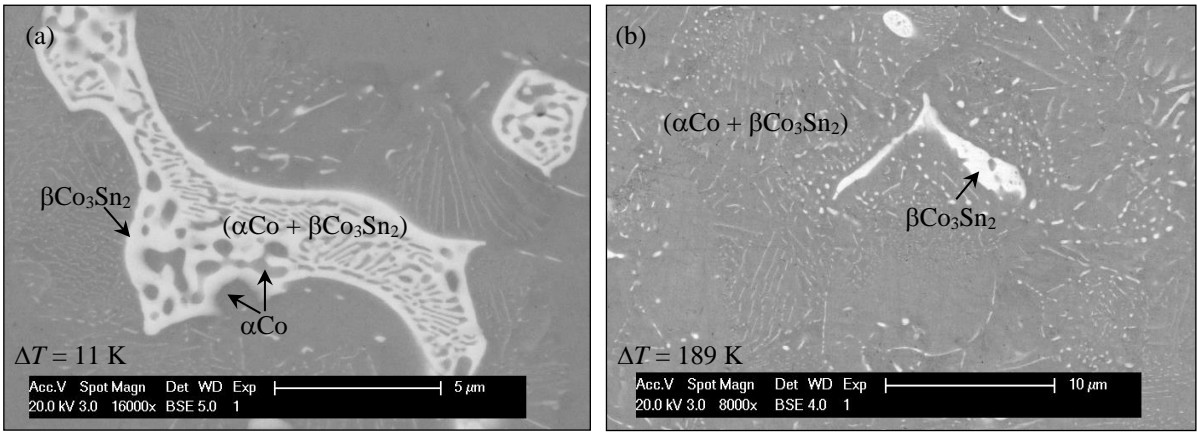

**Figure 3.** Eutectic structures within interdendritic spacings, (**a**) $\Delta T$ = 11 K, and (**b**) $\Delta T$ = 189 K.

### 3.2. Dendritic Growth Kinetics

The dendritic growth velocity is an important feature under the rapid solidification condition, which has an influence on the microstructure and physical properties of binary Co-4.54%Sn alloy. Figure 4a displays the measured and calculated growth velocity of the primary αCo phase, which presents two characteristics. At first, the dendritic growth velocity of primary αCo phase increased with the increase of undercooling, and a maximum velocity of 0.95 m/s was obtained at 175 K undercooling, which demonstrates a sluggish tendency for this alloy. However, the dendritic growth velocity slowed down once the undercooling exceeded 175 K and the dendrite decreased to 0.85 m/s growth velocity at the largest undercooling of 208 K. A Gaussian function was derived by fitting the actual dendrite growth versus undercooling to

$$V = 1.37 \times 10^3 \exp\left(-\frac{1.71 \times 10^{-21}}{k_B \cdot \Delta T}\right) \exp\left(-\frac{-1.14 \times 10^{-19}}{k_B \cdot T}\right) \tag{1}$$

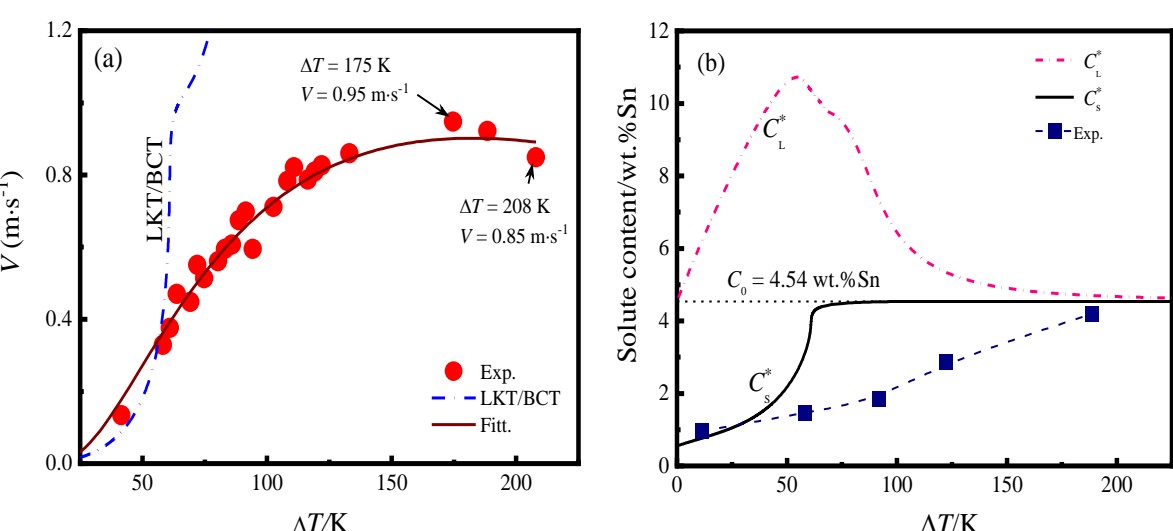

**Figure 4.** Measured and calculated dendritic growth velocity and solute content of rapidly solidified binary. Co-4.54%Sn alloy versus undercoolings, (**a**) dendritic growth velocity, and (**b**) solute content.

The LKT model for dendritic growth developed by Lipton, Kurz and Trivedi [20], which was further extended by Boettinger, Coriell and Trivedi (BCT) [21], has proven to be the most competent model for predicting dendritic growth characteristics with linear liquidus and solidus lines during rapid solidification. Cao et al. [22] modified the LKT/BCT dendritic growth theory to be applicable to predicting the kinetic characteristics of dendritic

growth in the alloy systems which have extremely curved liquidus and solidus lines. This model is comprised by the following seven principal equations:

$$\Delta T = \Delta T_t + \Delta T_c + \Delta T_r + \Delta T_k \tag{2}$$

$$R = \frac{\Gamma}{\sigma^*}\left[\frac{\Delta H}{C_{PL}}P_t\xi_t - \frac{2m'_L C_0(1-k_V)P_c}{1-(1-k_v)I_v(P_c)\xi_c}\right]^{-1} \tag{3}$$

$\Delta T$: bulk undercooling; $\Delta T_t$: thermal undercooling; $\Delta T_c$: solutal undercooling; $\Delta T_r$: curvature undercooling; $\Delta T_k$: kinetic undercooling; $V$: dendritic growth velocity; $R$: the dendrite tip radius; $\Delta H$: heat of fusion; $C_{PL}$: specific heat of the liquid phase; $P_t$ and $P_c$: thermal and solutal Peclet numbers; $\xi_t$ and $\xi_c$: thermal and solutal stability functions; $m'_L$: actual liquidus slope; $C_0$: alloy composition; $k_v$: actual solute partition coefficient; $I_v(P_c)$: solutal Ivantsov function; $\Gamma$: Gibbs–Thomson coefficient; and $\sigma^*$: the stability constant equal to $1/(4\pi^2)$.

From the LKT/BCT dendritic growth model, it can be seen that the dendritic growth velocity increased remarkably when the undercooling was larger than 25 K, and was 1~3 orders of magnitude larger than the actual dendritic growth velocity at the undercooling of 58~208 K. Therefore, the actual dendritic growth velocity was quite sluggish under the rapid solidification. According to the theoretical calculation of the LKT/BCT model, the dendritic growth is mainly controlled by the solutal diffusion if the undercooling is smaller than 91 K, which corresponds with the slower velocity. Subsequently, the effect of thermal diffusion becomes an increasingly important factor and finally replaces solutal diffusion as the dominant controlling factor once the undercooling exceeds 91 K.

The solute content of primary αCo phase was detected using the EDS method, and the results show the increasing tendency of undercooling, as seen in Figure 4b. For example, the solute content $C_S^*$ was 0.96 wt.%Sn when the undercooling $\Delta T$ was 11 K, which is the smallest level of undercooling acquired by the experiment. If the undercooling $\Delta T$ attained the largest undercooling 189 K of the experimental sample, the solute content also achieved 4.19 wt.%. According to the equilibrium diagram of Figure 1a,b, the solute content of primary αCo phase was only 0.56 wt.%Sn once the temperature decreased to 1760 K and the primary αCo phase began to grow from the liquid phase. Obviously, the solute trapping takes place during the rapid solidification. From Figure 2, it can be seen that the dendritic structure of the primary αCo phase refined remarkably with the increase of undercooling caused by the strong recalescence, the volume fraction of the βCo₃Sn₂ phase decreased apparently at the same time and a large number of Sn solutes diffused in the solvent interstitial of the primary αCo phase and led to the higher solute content. The variations of the liquid concentration $C_L^*$ and the solid concentration $C_S^*$ at the liquid/solid interface with undercooling were calculated using the LKT/BCT model, as illustrated in Figure 4b; the experimental value was close to the calculated $C_S^*$ result at the largest undercooling of 189 K. Apparently, the segregationless solidification may occur once the undercooling exceeds 189 K for binary Co-4.54%Sn alloy at t rapid solidification.

### 3.3. Microhardness and Electrical Resistivity

The microhardness of the binary Co-4.54%Sn alloy is illustrated in Figure 5a. As the undercooling was small, $\Delta T$ = 11 K, the microhardness $H_v$ was 228.6 HV, and then increased with undercooling. The maximum microhardness attained was 335.5 HV where the undercooling $\Delta T$ was 189 K, which is 1.47 times larger than the value of 11 K undercooling. The liner fitting function at different undercoolings is demonstrated as:

$$H_v = 221.13 + 0.65\Delta T \tag{4}$$

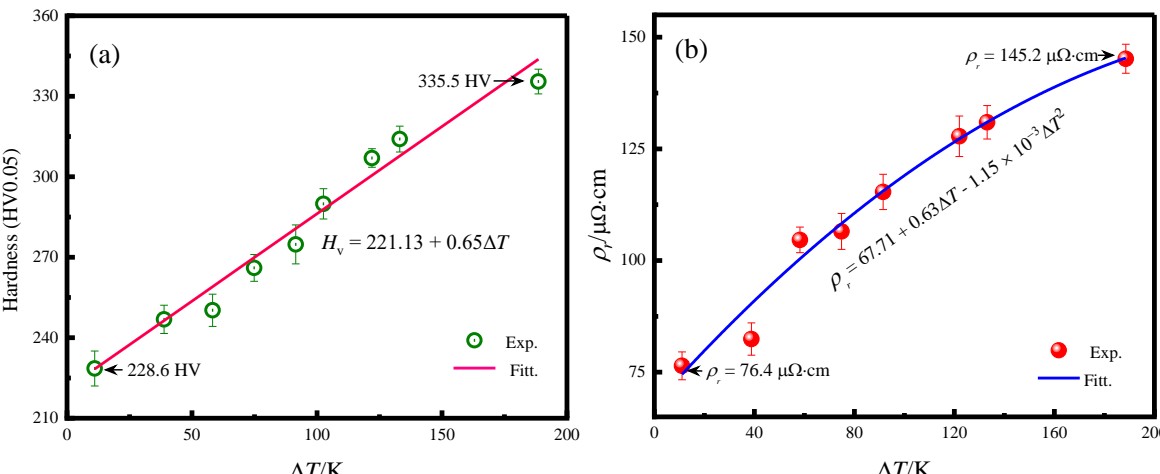

**Figure 5.** Microhardness and resistivity of Co-4.54%Sn alloy at different undercoolings, (**a**) microhardness and (**b**) resistivity.

Apparently, undercooling has a significant influence on microhardness. From the analysis above, it can be seen that large undercooling had more obvious recalescence, which led to the refined microstructure, and also enhanced the dendritic growth velocity and solute content. Therefore, the refined structure had increased grain boundary and larger solute content, which improved the microhardness under rapid solidification [23]. In addition, the microstructure also affected the hardness of Co-4.54%Sn alloy. Similarly, the undercooling also prompted the resistivity of binary Co-4.54%Sn alloy to rise, as displayed in Figure 5b. To describe the relationship between resistivity $\rho_r$ and undercoolings $\Delta T$, an expression is proposed:

$$\rho_r = 67.71 + 0.63\Delta T - 1.15 \times 10^{-3}\Delta T^2 \tag{5}$$

At the small undercooling of 11 K, the electrical resistivity $\rho_r$ was only 76.4 μΩ·cm. The electrical resistivity grew rapidly with the increase of undercooling, and the largest electrical resistivity $\rho_r$ of 145.2 μΩ·cm was obtained at the undercooling of 189 K, which is 1.9 times larger than that of 11 K. There are two reasons for this phenomenon. According to the classical nucleation energy definition, the nucleation energy of melt varies inversely with the square of the undercooling. Clearly, the nucleation energy decreases rapidly with the increase of undercooling, which enhances the number of crystal nuclei and refines the microstructure obviously. The growing crystal boundary hinders the electronic transmission, resulting in the electrical resistivity increasing. On the other hand, the reciprocal of relaxation time for lattice scattering caused by the impurity defect is proportional to the content of impurity concentration. Since the lattice distortion will cause strong electron scattering and reduce the mobility of free electrons, the more serious the lattice distortion of the total solute matrix in the constituent phase, the higher the resistivity of the alloy [24]. As shown in Figure 4b, the solute content of the primary ($\alpha$Co) phase enhances with the increase of undercooling, which generates the lattice distortion influence significant on the electronic transmission, thus improving the electrical resistivity.

### 3.4. Magnetic Properties

The magnetic characteristics of the Co-4.54%Sn alloy were measured at different undercoolings, as demonstrated in Figure 6. The Co-4.54%Sn alloy shows typical soft magnetic features (Figure 6a). Figure 6b describes the process of domain growth and magnetization rotation under an external magnetic field. The magnetic domains parallel to the applied magnetic field will grow, and the remaining magnetic domains will shrink. With the increase of magnetic field $H$, $B$ increases rapidly, and magnetic domain growth takes the way of magnetic domain wall movement. When the external magnetic field is increased,

the magnetic rotation begins, and the slope of the curve of *B* relative to *H* becomes smaller. Finally, the magnetic moment direction of each magnetic domain remains horizontal with the direction of the external magnetic field, and the *B* value does not change.

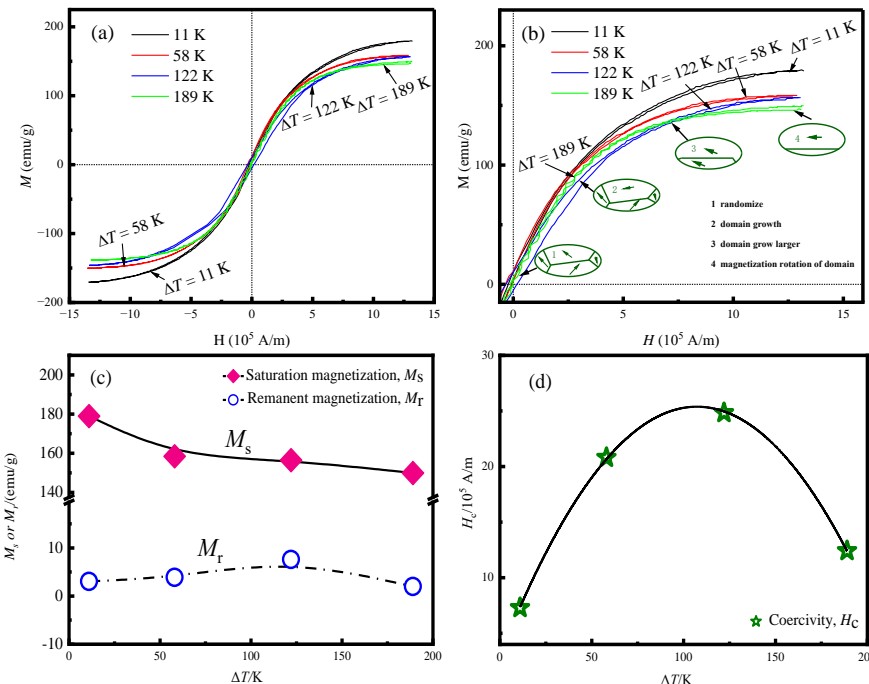

**Figure 6.** Magnetic characteristics of Co-4.54%Sn alloy solidified at different undercoolings. (**a**) Hysteresis loop, (**b**) magnetic domain growth and rotation process, (**c**) Ms and Mr and (**d**) coercivity Hc.

The saturation magnetization is generally related to the composition, the magnetic domain wall and the lattice constant of the material. The magnetic properties of the material are determined by the exchange interaction between the electrons in the material [25–27]. Its coercivity and saturation magnetization mainly depend on the material composition, crystal defect, internal stress and grain size [28]. Table 1 specifies the magnetic parameters of the Co-4.54%Sn alloys at different undercoolings.

**Table 1.** The magnetic property parameters of Co-4.54%Sn alloy under different undercoolings.

| $\Delta T$ | $M_s$ (emu/g) | $M_r$ (emu/g) | $H_c$ (kA/m) | $M_r/M_s$ |
|---|---|---|---|---|
| 11K | 178.92 | 3.04 | 7.3 | 0.017 |
| 58K | 158.40 | 3.91 | 20.81 | 0.025 |
| 122K | 156.49 | 7.63 | 24.84 | 0.049 |
| 189K | 149.94 | 2.01 | 12.42 | 0.013 |

With the increase of undercooling, the saturation magnetization of the alloy decreases from 178.92 emu/g to 149.94 emu/g, which is caused by the refinement of the microstructure, and the enhancement of dendritic growth velocity and solute content for the primary phase. Han et al. showed that alloy variants with larger average particle size usually have higher saturation magnetization, which is consistent with this result [29]. Compared with the undercooling of 11 K (3.04 emu/g, 7.3 kA/m), the undercooling of 189 K has lower remanent magnetization and higher coercivity, which are 2.01 emu/g and 12.42 kA/m, respectively. In addition, with the increase of undercooling, the squareness ratio showed an "increase first and then decrease" trend. The remanent magnetization and coercivity also exhibited a trend of increasing first, then decreasing with the increase of undercooling and solute content, as shown in Figure 6c,d. The reason is that the crystal particle and boundary are raised significantly due to the refinement of microstructure and enhancement of solute

content with the increase of undercooling, and a large amount of solute is distributed into the primary phase, which leads to the decrease of saturation magnetization and the increase of the remanent magnetization and coercivity. When the undercooling $\Delta T$ was 189 K, the remanent magnetization and coercivity were decreased in that the volume fraction of eutectic cell was higher than the undercooling of 11 K. Furthermore, a more in-depth study of the magnetic mechanism is still needed.

## 4. Conclusions

In summary, the broad temperature range of binary Co-4.54%Sn alloy was rapidly solidified using the fluxing method and the maximum undercooling attained 208 K. XRD analyses show that three phases of αCo, βCo$_3$Sn$_2$ and εCo were displayed during the solidification, which corresponds with the equilibrium phase diagram. The microstructure of the primary αCo phase remarkably refines and the volume fraction of βCo$_3$Sn$_2$ phase of interdendritic structure decreases evidently with undercooling. The dendritic growth velocity presents an increasing tendency at the beginning stage and the maximum velocity reaches 0.95 m/s when the undercooling is lower than 175 K, and then it decreases once the undercooling exceeds 175 K. Otherwise, the microhardness and resistivity of Co-4.54%Sn alloy increase with undercooling owing to the refinement of the microstructure, and the increase of solute content in the primary αCo phase is caused by the high undercooling and rapid solidification. In addition, the saturation magnetization reduces with the increase of undercooling, which is also due to the crystal particle and boundary increasing significantly and solute content being enhanced in the primary αCo phase, which results in the undercooling and dendritic growth velocity under rapid solidification. Moreover, the remanent magnetization and coercivity enhance shows a "increase first and then decrease" trend.

**Author Contributions:** Methodology, A.W.; software, W.W. and W.L.; formal analysis, W.W. and W.L.; data curation, W.W.; writing—original draft preparation, W.W. and W.L.; writing—review and editing, W.W. and W.L. All authors have read and agreed to the published version of the manuscript.

**Funding:** Financial support was provided by the National Natural Science Foundation of China [grant numbers 51931005, 52171048, 51571163] and the Key Industry Innovation Chain Project of Shaanxi Province (2020ZDLGY12-02).

**Data Availability Statement:** Not applicable.

**Conflicts of Interest:** The authors declare no conflict of interest.

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
