# Peer review of "Dendritic Solidification and Physical Properties of Co-4.54%Sn Alloy with Broad Mushy Zone"

_metals, doi:10.3390/met13061046_

Round 1

Reviewer 1 Report

The content is relevant and interesting.

The manuscript needs to be reworked. The language needs to be corrected. The right words have to be chosen. The Units should be checked.

Please check the units, especially in figure 5 and 6. Hardness in kg/m²? Are the units of magnetization right?

“of 11 K (15.69 A.m2.kg-1, 5.43-A-m-1), the” under table 1; wrong value, when I compare it with table 1.

Language (exemplified by the first paragraph):

The broad solidification temperature range is that the temperature range of solid-liquid two-phase mixture is larger than 0.2 TL, the alloy melt is in the semisolid state for this liquid/solid range during the solidification, this range is called broad mushy zone[1,2].“

·         “Broad” is used multiple times in the first paragraph at times it seems wrong. So either you mean “broadly” or really something with a distinct name?

·         TL is the liquidus temperature; please introduce it, put the L into the subscript and not just as a smaller letter.

·         “Alloy melt” is not semisolid. I get what you want to say but it is not right.

·         liquid/solid range = solidification interval?

“In terms of binary single phase alloy, a solid body nucleates from the liquid 28 phase to solid over a broad temperature interval, then it grows rapidly and forms the 29 single phase microstructure for a considerable time in this mushy zone[2].”

·         “In terms” à “In case”/”For”

·         “a solid body” à “solidification”

·         “single phase” à “single-phased”

·         for a considerable time in this mushy zone” I don’t understand what this part of the sentence is supposed to mean

“Therefore, the phenomenon of the solidification shrinkage and hot tear is caused by the thermal contraction, the higher density of the solid driving the cracking process and hindered liquid flow which prevents any void from being filled with liquid, and the several defects, for example, poor fluidity, shrinkage porosity, heat crack, segregation, and so on, take place inside this range[1-5].”

·         “hot tear” I think you mean the process not a tear by itself I would use “hot tearing”

·         the higher density of the solid driving the cracking process and hindered liquid flow which prevents any void from being filled with liquid” check grammar

·         “prevents any void from being filled with liquid” the void form because the shrinkage isn’t compensated by a fluid flow into the region.

·         “poor fluidity” à “low fluidity” or maybe “high viscosity”; the property of a fluid doesn’t appear to me as “defective”

·         “heat crack” do you mean hot crack or hot tear, or is it something else?

·         “and so on”

·         “range” à “interval”

The rapid solidification of high undercooling is an effective method to avoid or eliminate these defects.”

·         “The rapid solidification of high undercooling” the material is solidifying not the undercooling

Reviewer 2 Report

It is a scientific work about the dendritic solidification of a Co rich Co-Sn alloy with broad mushy zone.

The XRD analysis should be improved. The authors should determine the crystalline size, the microstrain, the lattice parameters and the percentage of each phase. There are linear methods as the Williamson-Hall method. One option is Rietveld analysis of the whole diffraction pattern. There are free software as Jana or Maud.

Figure 4 left: I suggest additional measurements in the 20 to 50 ΔT region in order to check the tendency in this region of the graph.

 Magnetism: The authors can calculate squareness ratio. Furthermore, the magnetic permeability is the initial or the maximum permeability?

Magnetism: Data are obtained only at two undercoolings. I suggest additional experiment to measure the magnetic response as a function of the undercooling.

Reviewer 3 Report

This manuscript studies the solidification and properties of Co-4.54%Sn alloy, which has a broad solidification range. The introduction is concise, well-written, and informative. The experimental procedure is explained clearly. The results are interesting and valuable. The discussion is sound. Therefore, I suggest this manuscript be published in this journal. Just a few minor changes are required that I hope the authors consider before the final version is submitted:

1. The English of the paper, although not bad, is heavy and requires much effort for understanding. I suggest the authors have it edited by a professional. Here are a few examples:

- Line 101: "with the matrix manner": manner does not make sense here. 

- Line 102: Co3Sn2 phase is better to be Co3Sn2 particles.

- Line 103: "distributes randomly into..." into should be in

- Line 105: whole is not a meaningful and suitable word in this sentence. 

these are a few suggestions for only one paragraph. 

2. Figure 1b caption for (b): xrd patterns obtained from the samples with different undercooling, (or a phrase like this that shows at a glance what is different between the spectra. 

3. This manuscript has not introduced the LKT/BCT dendritic growth model. What does LKT/BCT stand for? Reference is also needed.

4. As the authors know, data on a graph should always have error bars on them. No error bars are put on the data in this manuscript which is a shame. However, the calculation of error for the microhardness test is the easiest of all. therefore, I suggest the authors put error bars for all the graphs or at least for microhardness.

Round 2

Reviewer 2 Report

The quality and soudness of the manuscript has been improved.

The authors take into account the referee comments in the revised version.